# Toward an Integrated Consideration of 24 h Movement Guidelines and Nutritional Recommendations

**DOI:** 10.3390/nu15092109

**Published:** 2023-04-27

**Authors:** Elora Fournier, Edyta Łuszczki, Laurie Isacco, Emilie Chanséaume-Bussiere, Céline Gryson, Claire Chambrier, Vicky Drapeau, Jean-Philippe Chaput, David Thivel

**Affiliations:** 1Laboratory of the Metabolic Adaptations to Exercise under Physiological and Pathological Conditions (AME2P), Clermont Auvergne University, CRNH Auvergne, 63000 Clermont-Ferrand, France; 2Institute of Health Sciences, Medical College of Rzeszów University, 35-310 Rzeszów, Poland; 3Nutrifizz, 63000 Clermont-Ferrand, France; 4Aprifel, 75017 Paris, France; 5Department of Physical Education, Faculty of Education, Université Laval, Quebec, QC G1V 0A6, Canada; 6Healthy Active Living and Obesity Research Group, Children’s Hospital of Eastern Ontario Research Institute, Ottawa, ON K1H 5B2, Canada

**Keywords:** children, adolescents, adults, 24 h guidelines, movement behaviors, eating habits

## Abstract

While physical activity, sleep and sedentary behaviors are almost always considered independently, they should be considered as integrated human behaviors. The 24 h Movement approach proposes a concomitant consideration of these behaviors to promote overall health. Not only do these behaviors impact energy expenditure, but they have also been shown to separately impact energy intake, which should be further explored when considering the entire integration of these movement behaviors under the 24 h movement approach. After an evaluation of the prevalence of meeting the 24 h Movement and dietary recommendations, this review summarizes the available evidence (using English publications indexed in PubMed/MEDLINE) regarding the association between the 24 h Movement Guidelines and eating habits. Altogether, the results clearly show the beneficial impact of promoting the 24 h guidelines simultaneously, highlighting that the higher the number of respected movement recommendations, the better eating behaviors in both children and adults. Importantly, our results point out the importance of emphasizing the need to reach sedentary guidelines for better eating habits. Movement and dietary behaviors appear closely related, and giving recommendations on one might impact the other. Combining the 24 h Movement with dietary Guidelines might be more efficient than promoting them separately in public health strategies.

## 1. Introduction

The intimate relationship between health, sleep, sedentary behaviors (SB), physical activity (PA) and eating habits is defined as “conscious, collective, and repetitive behaviors, which lead people to select, consume, and use certain foods or diets, in response to social and cultural influences” [1], has already been highlighted in numerous studies. During the first years of life, sleep patterns evolve and are essential for efficient physical and cognitive development [2]. Sleep deprivation has been associated with poor eating habits, lower school performance and obesity in children and adolescents [3,4,5]. During adulthood, short sleep duration (i.e., less than 7 h per night) has been associated with mortality, type 2 diabetes, hypertension, cardiovascular diseases, cognitive disorders and obesity [6,7]. While sleep and appetite both have a diurnal rhythm, they seem closely associated and impact each other [5]. Poor sleep is associated with a dysregulation of appetite control, a decrease in PA and an increase in SBs such as screen use, which may ultimately lead to weight gain [5,8,9]. Being overweight and obese may also increase the likelihood of developing sleep disorders such as obstructive sleep apnea and further altering sleep quality [10]. In addition, while unhealthy diets seem to negatively influence the quality of sleep [10,11,12], insufficient sleep has been associated with higher fast-food consumption and sweet consumption in recent studies [8,13]. SBs such as screen time, especially TV viewing, have also been positively associated with sleep disorders [14,15] and obesity [16,17,18]. An increase in PA associated with a decrease in SB led to a better quality of life related to health [19,20,21]. The quality and behavior of the diet, PA, and SB are also interdependent and directly involved in the health and quality of life of young people [19,22] and adults [23,24]. Together, these studies demonstrate a deep connection between sleep, SB, screen time, PA and dietary habits. All these parameters are involved in a complex feedback loop that controls body homeostasis and health. For a long time, adult and youth interventions focused on a single parameter (e.g., increasing physical activity) but led to mixed results in the overall health of individuals [25,26,27,28].

The Canadian 24 h Movement Guidelines emerged as the first holistic recommendations combining PA, sleep and SBs adapted to each life stage. For children and youth (5–17 years), a ‘healthy’ 24 h includes an accumulation of at least 60 min per day of moderate to vigorous PA (MVPA), with muscle and bone strengthening activities at least 3 days a week, no more than 2 h per day of recreational screen time and limited sitting for extended periods [29]. Regarding the pattern of sleep, the guidelines recommend uninterrupted 9 to 11 h of sleep per night for those aged 5 to 13 years and 8 to 10 h per night for those aged 14 to 17 years, with consistent bed and wake-up times. For adults (18–64 years), the guidelines recommend between 7 and 9 h of sleep per day, no more than 8 h of SBs, including no more than 3 h of recreational SB daily, several hours of light physical activities each day, and an accumulation of at least 150 min of MVPA per week and muscle resistance training at least twice a week [30]. Since then, all the primary national and international organizations have been proposing this approach as part of their guidelines, all focusing on the idea that whatever the exact selected thresholds, it is recommended to reach them for as many movement behaviors as possible. The present paper will not be discussing the differences between guidelines and used cut-offs but will consider this general integrated 24 h approach whether the guidelines used by studies.

While most young people [31] and adults [32,33] fail to meet the 24 h Movement Guidelines, their adoption has been associated with numerous physical, mental and social health benefits [34,35,36,37]. While each of the behaviors composing these 24 h movement guidelines (PA, Sleep and SB) have also been individually associated with eating habits and the control of appetite, it seems today of interest to question how and to wish to extend their integration (under the 24 h movement approach) might also be associated with eating behaviors. The purpose of this study was then to analyze data exploring the association between adherence to the 24 h Movement Guidelines and dietary habits in both children and adults, considering the particular effects of the COVID-19 lockdowns and circumstances.

## 2. Methods

This narrative review was conducted by using PubMed/MEDLINE, ScienceDirect and MEDLINE databases, from November 2022 to February 2023, for English language meta-analyses, systematic reviews, randomized clinical trials, and observational studies from all over the world. The websites of scientific organizations, such as the WHO, were also searched. Five main topics were defined: (1) the 24 h Movement Guidelines among children and adolescents (from 3 to 18 years old); (2) the 24 h Movement Guidelines among adults (from 18 to 65 years); (3) prevalence of meeting dietary recommendation and associated effects in children and adolescents; (4) prevalence of meeting dietary recommendation and associated effects in adults; (5) interaction between the 24 h Movement Guidelines and eating habits in children, adolescents and adults. The principal search terms used were: 24 h movement guidelines, movement behavior, physical activity, sleep, sedentary behaviors, sedentary time, diet, eating habits, dietary guidelines, dietary recommendations, adult, youth, children, and adolescent.

Inclusion criteria: articles in English; eating habits and SBs issues; 24 h Movement Guidelines analysis; meta-analyses; systematic reviews; randomized clinical trials; observational studies; historical data. Exclusion criteria: articles in a language other than English, in vitro studies, animal studies, books, and narrative reviews.

## 3. Prevalence of Meeting the 24 h Movement Guidelines and Associated Effects

### 3.1. The 24 h Movement Guidelines among Children and Adolescents

Most young people do not meet the 24 h Movement Guidelines, particularly adolescents, girls and those from countries with a lower human development index. A recent systematic review and meta-analysis, conducted on preschoolers and adolescents (*n* = 387,437) from 23 countries reported that only 7.1% of youth met all three 24 h Movement Guidelines, while 19.2% did not meet any of them [31]. While growing up, children transitioning from primary to secondary school seem to experience unfavorable changes characterized by a decreased in sleeping time and PA and an increased time in SBs associated with screen time use [38]. Numerous studies reported the prevalence of meeting the 24 h Movement Guidelines in their countries [39,40,41,42,43]. However, data recorded from self- and proxy-reported questionnaires measuring 24 h movement behaviors in children and adolescents are difficult to compare from one study to another due to major content and reliability divergences [44]. Moreover, most studies included PA and SBs but less frequently sleep [45].

Lifestyle behaviors characterized by low PA levels, high SB levels and insufficient sleep may lead to numerous health issues. Moitra and colleagues showed that not meeting the guidelines for PA, sleep and screen time was associated with more aggravated obesity health-related problems in adolescents with obesity [46]. In contrast, meeting the 24 h Movement Guidelines has been associated with numerous health benefits on physical health (e.g., lower Body Mass Index (BMI), adiposity, cardiometabolic risk, higher cardiorespiratory fitness, better quality of life) and cognitive development (e.g., academic achievement) [36,37,47,48]. In addition, meeting the 24 h Movement Guidelines was associated with lower odds of being bullied or bullying others in children and adolescents [49,50], having an Internet addiction [51], and better mental health but also with lower severity of mental illness [52].

Since the beginning of 2020, the coronavirus disease 2019 (COVID-19) caused by the severe acute respiratory syndrome coronavirus 2 (SARS-CoV-2) has changed the world. The COVID-19 lockdown has led to a marked decline in adherence to 24 h Movement Guidelines among children and adolescents, even if major disparities were observed [53,54]. Children who attended school in person during the pandemic were more likely to meet the 24 h Movement Guidelines, especially screen time and MVPA, in contrast to sleep [55]. The decrease in movement behaviors observed during the COVID-19 lockdown has changed and implemented new habits in youth that were kept even at the end of the lockdown. After one year of the pandemic, the adherence to the 24 h Movement Guidelines was even lower (0.2% vs. 3.3%) than before the pandemic among 844 Spanish students [56]. The adherence to none of the recommendations had drastically increased from 28.9% before the pandemic to 66.3% after one year of the pandemic.

### 3.2. The 24 h Movement Guidelines among Adults

Establishing good habits during childhood may pay dividends later in life. Meeting the 24 h movement behaviors during adolescence appears to be related to a lower risk of abdominal obesity [57], type 2 diabetes [58] and depression [59] during adulthood compared to those who did not meet any of them. Early identification and recalibration of poor 24 h movement behaviors seem important because they can have long-term adverse health consequences.

However, no international survey has been carried out yet to assess the prevalence of meeting the 24 h Movement Guidelines among adults. The UK biobank cohort, established between 2007 and 2010, was explored to investigate those behaviors simultaneously in 398,984 adults classified according to their BMI [60]. Poor sleep, high TV viewing and low levels of PA, individually or combined, were associated with overweight and obesity in adults.

In 2022, a Canadian survey conducted among 8297 adults reported that only 7.1% of them met all three recommendations, while 19.1 % met none of them [35]. As observed in youth, meeting one, two or three guidelines was associated with better overall health in a dose-response gradient, i.e., that meeting three guidelines was better than meeting two, meeting two was better than meeting one, and meeting one was better than meeting none. This study also did not show any clear association between the respect of SB guidelines only and health, as shown in previous studies [61]. Future investigations should separately explore the relative contribution of various SB components (e.g., screen time vs. non-screen SB) and their associations with various health outcomes.

A multi-national cross-sectional study conducted among 2338 Latin American adults showed even worse results [33]. Only 1.6% of them met all the 24 h Movement Guidelines. Gender, education level and marital status seemed associated with guideline adherence. While fewer women tended to meet the MVPA recommendations, having a higher education level was associated with lower odds of meeting screen time guidelines. Finally, a 15-year longitudinal study reported that 21.3% of Thai adults (*n* = 167,577) met the 24 h Movement Guidelines with, contrary to the previous study, the lowest odds for males, those living in urban areas, unemployed and with low education level [32]. Interestingly, the follow-up spanned from 2001 to 2015 and showed a linear increase in the prevalence of meeting the SB guideline while meeting the sleep or MVPA guidelines declined since 2009. In Chilean adults (*n* = 2618), a national health survey conducted between 2016 and 2017 reported higher levels of individuals meeting all the recommendations, with 18.1% of adults meeting all three guidelines and 4.1% meeting none of them [62]. As shown in other studies, meeting none of the recommendations was associated with a higher probability of cardiometabolic disorders. However, comparisons between studies are difficult due to major divergences in terms of methodology (e.g., direct measurements with accelerometers vs. self-reports).

To investigate the possible associations between meeting movement guidelines and mortality risk, a survey conducted in American adults (*n* = 3471) in 2005–2006 (with an 11-year follow-up period) reported that only 12.3% of them met all recommendations while meeting recommendations was associated with lower mortality risk [34]. Of interest, reallocating time from SB to other movement behaviors (sleep or MVPA) was associated with lower mortality risk [63]. This result again reinforces the need to consider all recommendations and not each of them separately in public health guidelines.

During the COVID-19 pandemic, as observed among youth, adult movement behaviors changed drastically. Sedentary time increased at the expense of light PA or MVPA while sleep time remained stable, according to a longitudinal study based in the United States (*n* = 1992) [64]. The web-based Nutri-Net-Santé cohort, including 37,252 French adults, collected dietary and PA behaviors data during the COVID-19 lockdown in 2020 [65]. Findings show two distinct clusters characterized by unfavorable or favorable changes in movement behaviors. Most of the participants decreased their PA (53%) and increased their SBs (63%). However, the opposite trends were also observed in some of them, characterized by an increased in PA (19%) and home cooking (40%). During the lockdown period in Scotland, a decrease in walking time (mean difference = −55.58 min/week) was observed, while sleep (mean difference = 0.28 h/day), sitting time (mean difference = 29.10 min/day), and MVPA (mean difference = 68.40 min/week) increased (*n* = 3230) [66]. Interestingly, the MVPA level remained higher after the lockdown period. Individuals who positively changed one behavior during this time were more likely to report a positive change in another and maintained or improved them further after the lockdown and especially among participants displaying initially poor levels of each movement behavior. The 24 h movement behaviors are interdependent, and one change may lead to a cascade of other beneficial movement behavior changes. For example, having a good night’s sleep can facilitate the adoption of an active lifestyle. While adherence to the 24 h Movement Guidelines improved health and well-being in many studies, dietary habits may also predict health outcomes.

## 4. Prevalence of Meeting Dietary Recommendations and Associated Effects

Diet plays a major role in health. It modulates the gut microbiota [67], sleep duration and quality [68], metabolic status [69] and plays a key role in cardiovascular diseases [70]. According to the World Health Organization, a healthy diet “includes high consumption of fresh FV, daily breakfast consumption and discourages routine consumption of nutrient-poor foods that are high in sugars, saturated fats, trans fats and salt”. Based on current evidence, the traditional Mediterranean-type diet, including an important amount of FV, is often recommended to reduce cardiovascular diseases and improve metabolic health and general well-being [70,71].

### 4.1. Dietary Recommendations in Children and Adolescents

Having a healthy diet is of utmost importance during childhood to establish good nutritional habits and sustain growth. However, 88% of countries all around the world face some form of malnutrition (undernutrition, nutrient deficiencies, overweight, obesity, or diet-related non-communicable diseases) [72,73]. In Europe, 10.3% of children consume snacks, and 9.4% consume soft drinks on a daily basis [74]. Although most of the children eat breakfast (78.5%), less than half of the children consume fruits (42.5%), and fewer than a quarter of them consume fresh vegetables (22.6%) on a daily basis. Knowing the main role of FV consumption in overall health and body weight management, increased FV consumption among youth is crucial to face the obesity epidemic [69,75,76]. Patterns of eating habits are drastically different between developing and developed countries. For example, in this study, daily fruit consumption ranged from 80.8% in San Marino to only 18.1% in Kyrgyzstan and 19.2% in Lithuania [74]. Concerning vegetables, three-quarters (74.3%) of the children in San Marino consumed fresh vegetables every day compared to 9.1% of the children in Spain. Regarding sweet snacks, daily consumption ranged from 0.4% in Ireland to 32.8% in Tajikistan. Furthermore, recent findings from the Health Behavior in School-aged Children (HBSC) report showed that two in three adolescents in Europe and Canada do not consume enough nutrients rich foods, such as FV, while one in four adolescents eat sweets once a day [77]. Finally, a review of adolescent dietary patterns also highlighted an average consumption of FV below recommended values in almost all population studies [78].

Regarding the impact of eating habits on health, food-based dietary guidelines are essential to homogenize the discourse and represent interesting practical tools to promote healthier and sustainable eating habits. While dietary recommendations are highly diverse depending on countries (due to both nutritional and political specificities) and on the targeted life stage, they have been implemented in most of the regions so far [79]. Most of the time, dietary recommendations contain guidelines around FV consumption, fat, sugar, and sodium intake, and specific recommendations for protein sources according to the countries. Guidelines today continue to incorporate other components of eating habits, such as commensality, mealtime, and ultra-processed food recommendations based on the NOVA food classification system [80].

In line with the observations of the 24 h Movement Guidelines, eating habits have changed during the COVID-19 pandemic and associated lockdowns in children and adolescents. A systematic review highlighted both positive and negative results of eating habit changes in children and adolescents [81]. It reports an increase in home-cooked meals, fruits, vegetables, legumes, and bread, but also in snacks, French fries, sweets, and bakery products. In contrast, fast food and soft drink consumption decreased. While the general pattern among youth during the COVID-19 pandemic is in favor of healthier eating habits, youth from lower socioeconomic groups showed a tendency towards more unhealthy eating habits [82]. Therefore, the increased incidence of obesity in children and adolescents after the COVID-19 pandemic seems to be more related to a reduction in PA and an increase in SB (and screen time) than to the modification in dietary habits [83].

### 4.2. Dietary Recommendations in Adults

A global dietary quality assessment conducted in 185 countries from 1990 to 2018 using data from the Global Dietary Database project reported a mean international Alternative Healthy Eating Index score of 40.3 (ranging between 0 and 100 from the least to the most healthy) [84]. The quality of the diet increased slightly throughout the world (except in South Asia and Sub-Saharan Africa) and was higher in women compared to men and for more educated individuals. Major discrepancies remain between countries with the highest dietary pattern scores identified in low-income countries (South Asia and Sub-Saharan Africa), mainly due to consistently low consumption of red/processed meat, salt and sugar-sweetened beverages. In contrast, an increase in FV, legumes, nuts, and whole grain consumption over time in high-income countries has improved dietary quality despite high and stable consumption of red/processed meat, salt and sugar-sweetened beverages. The Eat–Lancet Commission on Food, Planet, and Health suggested common dietary guidelines for a healthy and environmentally sustainable diet, integrating limiting red meat, poultry (≤98 g/week of unprocessed red meat and processed meat combined), and egg (≤91 g/week) consumption, and moderate levels of fish (≤196 g/week) and dairy (≤250 g/day of milk, cheese, and yogurt combined) consumption [85]. It is noteworthy that food associated with adult health (fruits, vegetables, whole grain cereals, legumes, nuts, and fish) has lower environmental impacts (except fish) and should be encouraged in future policies [86].

Despite the improvement in diet quality in the European Union (EU) in 2019, 33% of the adult population did not consume any FV daily, while only 12% of people consumed the recommended five portions or more daily, according to the European Health Interview Survey [87]. Among EU member states, Ireland harbors the highest daily intake of five FV portions or more per day (33%), followed by the Netherlands (30%), Denmark (23%) and France (20%) [87]. In Japan (*n* = 88,527), plant and fish food consumption have decreased, contrary to bread and dairy consumption [88]. In Australia, none of the young adults met all the Australian Eating recommendations for grain food, vegetable, fruit, dairy, meat and discretionary foods, including energy-dense, nutrient-poor foods such as sweetened beverages [89]. Only 32% of young adults (*n* = 1005) met the fruit recommendations, while the majority (76%) exceeded the discretionary food recommendation and 81% saturated fat. Among Canadian adults (*n* = 15,512), only 23% consume FV more than five times a day [90]. In the USA, FV consumption has decreased over time. In 2009, less than 25% of adults consumed at least five daily servings of FV [91] compared with 13.1% in 2013 [92] and only 12.3% and 10.0% for FV respectively in 2019 [93]. Only 30% and 40% of Filipino working adults (*n* = 1264) respected the recommended intake of FV and based the majority of their energy intake on fats, proteins, and poor nutrient-dense foods [94]. In Norway, 31% of women and 17% of men met the five-day recommendation for FV (*n* = 11,425) [95] while in Brazil, only 38% of women (*n* = 930) and 24% of men met the minimum recommendation for FV intake [96].

During the COVID-19 pandemic and the associated lockdowns, significant modifications in food consumption and practices were observed across the globe. In Ireland, Great Britain, the United States and New Zealand (*n* = 2360), the inhabitants consumed fewer takeaway foods, baked more and generated less food waste [97]. Regarding diet quality, an increase in FV consumption was reported in Ireland, Great Britain, the United States, New Zealand [97], Brazil [98] and Malaysia (*n* = 1045) [99] while saturated fat intake increased. The web-based French NutriNet-Santé cohort cited above reported two different clusters characterized by unfavorable or favorable changes in their eating habits during the COVID-19 lockdown in 2020. Most of the participants reported an increase in snacking and in sweets, cookies, and cake consumption, followed by a decrease in FV and fish consumption [65]. In contrast, a second cluster reported working on balancing their diet and increased FV consumption while decreasing sandwich, pizza, sweets, cookies, cake or alcohol consumption. Similar results were also observed in the NutriQuébec cohort, where improvement in diet quality and a decrease in the prevalence of food insecurity was observed in Canadian adults during the COVID-19-related early lockdown [100]. In Malaysia, the eating pattern during the pandemic was also followed by an increase in fat and cereal consumption associated with weight gain. In Brazil (*n* = 1929), the increase in FV consumption was followed by a concomitant increase in sweet and fried food consumption [98]. Thus, modifications to our SB, PA and sleep behaviors during the COVID-19 lockdown were also associated with variations in our eating habits that may be correlated.

Furthermore, improved diet quality, assessed with the Alternate Healthy Eating Index score or the Alternate Mediterranean Diet score, was associated with a lower risk of death from any cause [101]. Despite knowing the link between diet, well-being and health, few adults meet current guidelines. Likewise, understanding what drives food consumption is important to predict and favor healthier eating habits.

## 5. Interactions between the 24 h Movement Guidelines and Eating Habits

Long considered independent of each other, movement behaviors and eating habits are, in fact, closely related. According to Taylor’s work presented in one of his personal communications and which has been supported on several occasions afterward, the regulation of individuals’ energy intake and, more broadly, of their overall appetite control is highly influenced by their activity and movement behaviors [102,103,104]. Briefly, Taylor et al. proposed in 1974 that human daily movement behaviors (and mainly physical activity level) and their energy intake were associated following a J-shaped curve. Briefly, the higher individuals’ physical activity level, the better the regulation of their appetite control. However, according to this J-shaped curve, physically inactive people are situated in a non-regulated zone when it comes to their energy intake and appetite control. Since then, the scientific literature has been proposing strong experience-based evidence supporting this theory, both in adults and youth [103,105]. Above physical activity/inactivity itself, sleep deprivation also leads to a modification of eating habits [8]. Nevertheless, most of the time, experimental studies focus on one component at a time (e.g., effect of eating patterns on sleep or SB or PA) and do not evaluate potential synergies among them [5,10,15,20]. With the 24 h movement continuum approach, it seems interesting to study all these parameters in synergy. In fact, there is growing scientific and clinical interest in the potential links between eating and movement behaviors in children and adolescents (Table 1), but no studies have been conducted in adults yet. It seems indeed necessary to better understand the possible interactions between these behaviors in order to optimize our interventions and advice and also to avoid potential compensatory responses in both youth and adults [106]. Changing behavior can lead to a domino effect whereby other healthy changes can follow. This clustering of healthy behaviors is well documented in the literature. Similarly, unhealthy behaviors also tend to cluster together, and it may become very challenging to escape this vicious circle.

A cross-sectional study conducted between 2011 and 2013 on 5873 children from 12 countries examined the association between 24 h movement behaviors and eating habits in children [107]. Movement behaviors were measured objectively using accelerometry, while the eating habit was described as “healthy” and “unhealthy” using dietary pattern scores derived from the statistical analysis of the food frequency questionnaire. Results highlighted an association between meeting more movement behavior recommendations with better dietary patterns (positive loadings for vegetables, fruits, whole grains, low-fat milk, etc.) similarly across sites and for boys and girls [107]. In particular, limiting screen time habits (and consequently SB) was the guideline most strongly associated with a healthier dietary pattern scores both alone and in combination with MVPA recommendations. Therefore, the SB recommendations seem to have a specific role in promoting healthier dietary patterns in children. In line with this study, a European cross-sectional study based on the HELENA cohort that collected data on 1448 adolescents from eight cities supported a strong association between PA and SB, such as screen time and dietary habits. They measured PA objectively by accelerometry and dietary patterns by 24 h dietary records. The findings showed that those who did not meet the PA and screen time recommendations were less likely to eat FV and more likely to have a higher intake of fats, oils, and sugar-sweetened beverages [108]. Altogether, the studies by Thivel and Moradell suggest that meeting the screen time recommendation (and consequently SB guidelines) was the behavior most strongly associated with healthier dietary patterns.

**Table 1 nutrients-15-02109-t001:** Studies evaluating the interactions between eating habits and movement behaviors in children and adolescents.

Authors	Date	Title	Country, N	Age	Analyses	Results
[109] Christofaro	2016	Higher screen time is associated with overweight, poor dietary habits and physical inactivity in Brazilian adolescents, mainly among girls	Londrina, Brazil 1231 students	14–17 years	Nutritional status (normal weight or overweight/obese): BMI, Eating habits, screen time & PA: questionnaire Associations between variables (PA levels, nutritional status, eating habits) assessed by binary logistic regression, adjusted for lifestyle and sociodemographic variables.	93.8% of boys and 87.2% of girls spent more than 2 h per day in screen-time activities among adolescents. The prevalence of overweight and physical inactivity tends to increase with increasing time spent on screen activities for both sexes after adjustments. Screen time of more than 4 h per day compared with less than 2 h per day was associated with low consumption of vegetables, physical inactivity and high consumption of sweets only in girls and the consumption of soft drinks in both sexes.
[110] Delfino	2018	Screen time by different devices in adolescents: association with physical inactivity domains and eating habits	Brazil 1011 adolescents	10–17 years	PA, screen time & dietary habits: questionnaires	70% of adolescents reported high use of mobile phone/tablet, 63% of TV or computer and 24% of videogames. Boys present higher use of video games than girls, while the high use of mobile phone/tablet was higher among girls. High use of TV, computer, video games, and snack consumption were associated. High use of computers was associated with physical inactivity and the consumption of fried foods. The mobile phone was associated with sweets consumption. Associations of clusters using screen devices with high consumption of sweets, snacks and fried foods even after controlling for confounding variables.
[111] Huo	2022	Screen Time and Its Association with Vegetables, Fruits, Snacks and Sugary Sweetened Beverages Intake among Chinese Preschool Children in Changsha, Hunan Province: A Cross-Sectional Study	Hunan Province (China) 1567 preschoolers	3–6 years	Cross-sectional study. Eating habits, screen time & PA: questionnaire. Covariates were parental feeding practices, children eating habits and food neophobia	The screen-time mean of preschoolers: 1.36 ± 1.26 h 54.3% of children spent more than one hour on screen devices. Children who spent longer screen time consumed vegetables and fruits less frequently, while having a higher consumption of sugary sweetened beverages and snacks. The association of screen time with sugary sweetened beverages and vegetables still remained significant after adjustment of parental feeding practices, sociodemographic confounders and children’s eating habits
[112] LeBlanc	2015	Correlates of objectively measured sedentary time and self-reported screen time in Canadian children	ISCOLE cohort—Canada 567 children	9–11 years of age	Cross-sectional study. MVPA & nightly sleep duration: 24 h waist-worn accelerometry. ST habits: self-report. Dietary patterns: food frequency questionnaire. Diet was described by two components derived from principal component analysis: “healthy” and “unhealthy” dietary pattern scores. Covariates included in the multilevel statistical models included sex, ethnicity, number of siblings, and household income.	Daily SED of children averaged 8.5 h. Girls and boys presented no differences in total screen time or total SED. More video game/computer usage was reported in boys than girls. A higher waist circumference and BMI z-scores were observed in boys compared to girls. SED and ST were both correlated with waist circumference and the number of TVs in the home. SED and sleep duration were negatively associated. ST was negatively associated with weekend breakfast consumption, healthy eating pattern score and positively associated with unhealthy eating pattern score, mother’s weight status and father’s education. Boys and girls presented few common correlates.
[113] Liu	2022	The Association of Soft Drink Consumption and the 24 h Movement Guidelines with Suicidality among Adolescents of the United States	USA 73,074 adolescents	15-years-old or above	Data extracted from Youth Risk Behavior Surveys (YRBS) collected from 2011 to 2019. PA, screen time, and dietary habits: questionnaires. Covariates included in the Binary logistic regression models included demographic factors, weight status, dietary behaviors & depressive symptoms	An increased risk of suicidal ideation and suicide plan in adolescents was associated with not meeting all the recommendations of the 24 h guidelines compared with those who meet all the recommendations. Consumption of soft drinks more than 3 times per day was associated with an increased risk of suicidality including suicide plan, attempt and ideation, as well as suicide attempt with medical treatment, regardless of sex or whether the recommendations of ST, PA, and sleep duration were met.
[114] Miguel Angel	2022	Is adherence to the 24 h Movement Guidelines associated with Mediterranean dietary patterns in adolescents?	Spain 1391 adolescents	11–16 years	Cross-sectional study. PA: Physical Activity Questionnaire. Recreational ST: Spanish version of the Youth Leisure Sedentary Behavior Questionnaire (YLSBQ). Sleep duration: Spanish translation of a self-reported sleep questionnaire. Adherence to the Mediterranean diet: Spanish version of the KIDMED questionnaire. Covariates included were age, sex, socioeconomic status, and body mass index (z-score).	Adolescents who met all three 24 h Movement Guidelines obtained higher adherence to the Mediterranean diet, were more likely to consume a FV once a day, consume fish regularly, and eat cereal or grains for breakfast and less likely to consume commercially baked goods or pastries for breakfast and to eat sweets and candies several times a day, than those who did not comply with the three 24 h Movement Guidelines.
[115] Miguel-Berges	2019	Combined Longitudinal Effect of Physical Activity and Screen Time on Food and Beverage Consumption in European Preschool Children: The ToyBox-Study	6 European countries 2321 children	3.5–6 years	Multicenter longitudinal ToyBox study from 2012 to 2013. Dietary habits: semiquantitative food frequency questionnaire. ST: parental questionnaire. PA: means of pedometers (except Belgium—accelerometers). Covariates included in the linear mixed effects model included *z*-BMI, maternal education, and intervention	More than half of the children (50.4%) did not meet the screen time and PA guidelines at both baseline and follow-up. 0.6% of children met both screen time and PA recommendations at both baseline and follow-up. Children meeting both screen time and PA recommendations consumed fewer sweets, fizzy drinks, juices, desserts, and salty snacks and consumed more water, FV, and dairy products than those who are not meeting both recommendations
[108] Moradell	2022	Are Physical Activity and Sedentary Screen Time Levels Associated With Food Consumption in European Adolescents? The HELENA Study	HELENA study—8 European cities 1448 adolescents	?	PA: objectively measured by accelerometry. Dietary intake: 24 h dietary records. Groups were made between adolescents according to PA and SST recommendations.	Intake of savory snacks was higher in groups who did not meet any of the recommendations in both sexes. Among males, those who met both PA and SST recommendations were more likely to drink or eat milk, yogurt, and water, while those who did not meet the recommendations were more likely to drink sugar-sweetened beverages. Among women, those who did not meet the recommendations were more likely to have a higher intake of fats and oils and less likely to eat FV.
[116] Rubín	2020	Prevalence and correlates of adherence to the combined movement guidelines among Czech children and adolescents	Czech Republic 355 children and 324 adolescents	3–13 years and 14–18 years	PA and sleep duration: accelerometers. Recreational ST: parent proxy-reported in children and self-reported in adolescents	Approximately 6.5% of children and 2.2% of adolescents met all recommendations of the combined movement guidelines. In children, children with overweight or obese fathers and girls had lower odds of adherence to both movement guidelines. If children reported regular FV intake, participated in organized PA, or if their fathers had a university degree, they had higher odds of meeting specific two-recommendation combinations. Lower odds of meeting specific combinations of recommendations was associated with paternal overweight and obesity and with high sleep efficiency. In adolescents, a correlation between meeting specific combinations of any two recommendations and sex, FV intake, organized PA, and active play was highlighted.
[117] Sampasa-Kanyinga	2022	Movement behaviors, breakfast consumption, and fruit and vegetable intake among adolescents	Ontario (Canada) 12,759 students	15.2 ± 1.8 years; 56% females	Cross-sectional study. Movement behaviors, screen time, sleep duration and eating habits: self-reported (questionnaire). Multivariable ordered logistic regression analyses were adjusted for age, sex, ethnoracial background, subjective socioeconomic status, and BMI *z*-score. Subjective social economic status was assessed using an adapted version of the MacArthur Scale for Subjective Social Status.	Respecting the PA, ST, and sleep recommendations is associated with more FV intake and more frequent breakfast consumption among adolescents. A dose–response gradient was observed between the number of recommendations met (3 > 2 > 1) and more frequent breakfast consumption and FV intake. The compliance with all three recommendations was the best combination. 21%, 24% and 33% of students met the PA, screen time and sleep duration recommendations, respectively. Only 3.6% of the students met the three recommendations, while 44.6% met none of them. More males comply with all three recommendations than females. Respecting the screen time or PA recommendation only was not associated with the frequency of breakfast consumption. Participants who complied with the screen time recommendation only were not associated with FV intake.
[118] Soltero	2021	Associations between Screen-Based Activities, Physical Activity, and Dietary Habits in Mexican Schoolchildren	Mexico 874 children	9.6 (±1.0) years	PA, screen time & dietary habits: School Physical Activity and Nutrition survey. Organized activity/sports participation, unhealthy dietary habits, and household income were correlated with screen-based activities.	Contrary to computer and video game use, TV watching was associated with decreased participation in organized activity/sports participation. Compared to girls, boys spend more time watching TV and playing video games. Among boys, all screen-based activities were associated with age In girls, video games and computer use were associated with higher income.
[107] Thivel	2019	Associations between meeting combinations of 24 h movement recommendations and dietary patterns of children: A 12-country study	ISCOLE cohort—12 countries 5873 children	9–11 years of age	Cross-sectional study. MVPA & nightly sleep duration: 24 h waist-worn accelerometry. ST habits: self-report. Dietary patterns: food frequency questionnaire. Diet was described by two components derived from the principal component analysis: “healthy” and “unhealthy” dietary pattern scores. Covariates included in the multilevel statistical models included age, sex, highest parental education, and BMI *z*-score.	Based on data collected between 2011 and 2013. The more movement behavior recommendations were met, the best dietary patterns were. The recommendation the most strongly associated with better dietary patterns was screen time. Similar associations were found for boys and girls and across study locations.

BMI—body mass index, FV—fruits and vegetables, MVPA—moderate-to-vigorous physical activity, PA—physical activity, SED—sedentary time, ST—screen time.

This observation was also reported specifically on a national scale. In Spain, a cross-sectional study conducted on 1391 adolescents (11–16 years) demonstrated that those who met all three 24 h Movement Guidelines obtained higher adherence to the Mediterranean diet and were more likely to consume FVs once a day than those who did not meet the three recommendations [114]. Christofaro’s study, conducted on students from Londrina in Brazil (14–17 years), showed a lower consumption of vegetables in both sexes associated with a screen time of more than 4 h per day compared with those respecting the recommendation of no more than 2 h per day [109]. Additionally, a cross-sectional study carried out on 1567 preschoolers (3 to 6 years) in China also reported a lower FV consumption and a higher consumption of snacks and sugar-sweetened beverages in children with longer screen time [111]. Again, a study conducted on 355 children (8–13 years) and 324 adolescents (14–18 years) from the Czech Republic highlights that regular FV intake was associated with higher odds of meeting the combination of PA and screen time in children and adolescents, and screen time and sleep recommendations in children only [116]. Screen time was also negatively associated with a healthy eating pattern score (including FV recommendations) in Canadian children with a mean age of 10 years (*n* = 567) [112]. Screen-based activities were also associated with unhealthy eating habits in a study conducted on Mexican schoolchildren [118]. Strong associations were again highlighted between high TV, computer, and video game use with the consumption of snacks, sweets, fried foods and SB among 1011 Brazilian adolescents [110].

A multicenter European study, ToyBox, involving 2321 children from six European countries during one year, investigated compliance with PA and screen time recommendations at two-time points in relation to food and beverage consumption [115]. PA was usually evaluated by means of pedometers assessing the step number per day, while screen time and diet information were collected via parental questionnaires. The results also showed a strong association between the meeting of the PA and screen time guidelines at the two-time points with healthy dietary habits (FV consumption and water intake) and lower consumption of energy-dense products (sodas, sweets, desserts, and salty snacks) despite the very low percentage of children who met both recommendations (0.6%). Similar observations were made in 12,759 Canadian students (15.2 ± 1.8 years), showing a dose–response gradient between the number of recommendations met (3 > 2 > 1) and FV intake, with compliance with the three recommendations being the best combination [117]. The median frequency of FV intake was identical between girls and boys. However, compliance with only the screen time recommendation was not associated with FV intake.

Current evidence highlights the strong relationship between movement and eating habits or diet quality. Adhering to the 24 h Movement Guidelines seems closely associated with a healthy diet, especially the SB recommendation. Previous observations in children suffering from obesity already showed the central role of SB as a health indicator even more than PA [119]. Indeed, as energy intake follows a J-shaped curve according to PA levels, it is in homeostasis, with an energy expenditure only above a certain threshold of PA [36]. Thus, low energy expenditure disrupts appetite control processes, leading to overconsumption. While previous models [120] clearly showed the importance of the PA level and SBs of children on their present health, but also on their future (when adult) activity and movement behaviors as well as health. Julian and collaborators recently updated this model by incorporating the overall 24 h movement approach and by pointing out the central role of its interactions with appetite control in these associations [36]. Another potential mechanism explaining the link between movement (especially SBs and sleep) and eating habits is through its impact on body size satisfaction. Accordingly, there is evidence highlighting that screen time is associated with body size dissatisfaction and the desire to reduce body size in adolescents [121,122] caused by exposition to appearance-highlighting content, including retouched images, leading to unrealistic body ideals dissatisfaction [123,124] which increase social comparison [125]. Body size dissatisfaction can, in turn, lead to the adoption of unhealthy dietary habits such as low FV consumption [126]. Altogether, the available evidence suggests that following the 24 h Movement Guidelines may improve appetite control, body size satisfaction and eating habits to ultimately have beneficial effects on health across the life cycle. However, further studies should focus on adults to evaluate these potential theoretical interactions.

## 6. Limitations and Future Directions

The review points out the predominance of cross-sectional studies, clearly signaling the need for further longitudinal and clinical studies in children, adolescents and adults dealing with the quality of their diet, sleep, SB and PA. Even if we made every effort to ensure that the collected studies were as objective as possible, as indicated by the large number of studies cited, the clear inclusion/exclusion criteria, and the reporting of all study results, a narrative review is a form of scientific work that may carry a greater risk of subjective evaluation compared to a systematic review or meta-analysis.

Furthermore, energy and nutritional deficiencies, which may also contribute to the risk of the non-communicable disease, would be useful to consider. In further studies, it is important to improve people’s awareness of the impact and importance of diet, sleep, SB and PA through health promotion interventions and programs.

## 7. Conclusions

Dietary and movement behaviors appear to be closely related and positively influence each other. While adherence to the 24 h Movement Guidelines or following dietary recommendations improve health and well-being independently in both adults and youth, interactions between these factors also play a central role. Giving recommendations on one of these parameters will inevitably impact the others. Consequently, a broader and integrated 24 h approach is necessary and will be more efficient than a segmented or siloed approach that only considers one component individually. Therefore, combining the 24 h movement recommendations with the dietary recommendations could produce a stronger and better impact on overall adult and youth health than having them separately. For example, late-night television watching is associated with increased snacking and poor sleep quality, while insufficient sleep is associated with increased food intake and increased SBs. Not considering recommendations around eating habits when providing advice on 24 h movement behaviors is misleading and ignores important connections that should be considered. Future studies should continue to investigate this all-encompassing 24 h approach to public health guidelines while keeping in mind that simple messaging is important for better uptake.

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
