# Peer review of "Toward an Integrated Consideration of 24 h Movement Guidelines and Nutritional Recommendations"

_nutrients, 2023, doi:10.3390/nu15092109_

Round 1
Reviewer 1 Report
Dear Authors
The article presented is very pertinent and interesting, mainly because it deals with a current issue - that of the increase in the number of people with obesity and excess weight. It is a problem that affects all ages.
The most interesting aspect for me is the proposal to add nutritional counseling to the "24 Hour movement guidelines". So it seems to me that the title could be a little more "cheeky" in referring to that need. The conclusions could also suggest that change.
Reviewer 2 Report
Abstract
The aim of this section should be to motivate the future reader to be interested in the content of the full paper and to be potentially citable in future work. However, what is presented is a very, very general and obvious summary of the current situation with respect to the topics of the article and this is not really motivating content, neither to get an idea of what the authors have done in their research, as we do not even know a preview of the main result detected, nor to find out what they have really done.
This section needs a major overhaul.
Introduction
Lines 35-67. Good review to approach the research problem.
Lines 72-73. We know that the journal template allows truncation of words at the end of lines, however, in many, many cases (such as this one at the end of line 72) there is a spelling mistake as the syllable is broken. Please check the entire article for such errors.
Methods
Ok, good explanation
3. Prevalence of meeting the 24-Hour Movement Guidelines and associated effects
3.1. The 24-Hour Movement Guidelines among children and adolescents
Lines 120-122. If you report that "Numerous studies reported the prevalence of meeting the 24-Hour Movement Guidelines in their countries" you should include the references here, or at least the most recent and comprehensive ones.
Please consider including this citation in this section: García Perujo, María, and Pedro José Carrillo López. 2020. "Niveles De Actividad física Y Calidad De La Dieta En Escolares De Educación Primaria". Revista Iberoamericana De Ciencias De La Actividad Física Y El Deporte 9 (2):16-31. https://doi.org/10.24310/riccafd.2020.v9i2.7155.
Thank you
3.2. The 24-Hour Movement Guidelines among adults
Ok, good and comprehensive explanation
4. Prevalence of meeting dietary recommendations and associated effects
Lines 227-229. Please use inverted commas '' and not the Germanic ' ('includes high consumption of fresh FV, daily breakfast consumption and discourages routine consumption of nutrient-poor foods that are high in sugars, saturated fats, trans fats and salt').
The rest is OK.
4.1. Dietary recommendations in children and adolescents
Ok, correct explanation.
4.2. Dietary recommendations in adults
Ok, correct explanation.
General comments on point 4 and subsections.
The explanations are extensive, correct and up to date, but they do not really go beyond the level of information and are very well known and obvious.
5. Interactions between the 24-Hour Movement Guidelines and eating habits
Line 359. What "experimental studies"? Please include references.
Lines 371-391. Please consider including this citation in this section: Ubago-Jiménez, J.L., R Chacón-Cuberos, P Puertas-Molero, and I.A. Ramírez-Granizo. 2020. "Influencia De La Dieta Y hábitos físico-Saludables En Escolares". Revista Iberoamericana De Ciencias De La Actividad Física Y El Deporte 9 (1):106-13. https://doi.org/10.24310/riccafd.2020.v9i1.8306.
Overall, good review, good examples, good data and well explained. Good work.
6. Limitations and Future Directions
Ok, good explanation
7. Conclusions
As a result of an obvious review there are some equally obvious conclusions. However, the conclusions can be considered to be supported by the results of the review and are correct.
Reviewer 3 Report
I think this review is trying to do too much in one narrative review and should instead try to identify a time period that they are intersted in (i.e. COVID-19 on, pre-COVID-19) and pick a specific part of the world. You are all over the place with this narrative review.
In your introduction you emphasize the relationship between sleep, diet, PA and SB however, you end your introduction by simply talking about the association between diet and PA. I really thought you would talk about the interaction of all 3. Can you please modify either your introduction to reflect what you will focus on or your paper to reflect your introduction.
What were your search terms?
How did you identify adults? What about older adults?
Why pick the Canadian guidelines for PA and not say ACSM? Or NIH? You need to justify why you picked the Canadian guidelines and then applied them to all other countries.
This narrative review needs significant work and a focus. I encourage the authors to create a focus of this narrative review instead of trying to cover everything at the same time. I think this could have significant value but you need to focus. Unless you want to do something along the lines of what Firth and colleagues (2020) have done.
Firth, J., Solmi, M., Wootton, R. E., Vancampfort, D., Schuch, F. B., Hoare, E., ... & Stubbs, B. (2020). A meta‐review of “lifestyle psychiatry”: the role of exercise, smoking, diet and sleep in the prevention and treatment of mental disorders. World Psychiatry, 19(3), 360-380.
Round 2
Reviewer 3 Report
I appreciate the authors addressing my concerns. The manuscript is much more readable and seems to have a consistent flow.
My only suggestion now is to address in your objective statement that you will be comparing pre, during the post-COVID-19 pandemic. Other than that I think this is great.
Author Response
We thank the reviewer for this new positive evaluation of our work. Although we do not perform here a strict comparison of the pre-per and post COVID context, we have, as requested by the reviewer, detailed in the objective of the paper that the particular effects of the COVID-19 periods have been considered and discussed.